# Activation of Peripheral Cannabinoid Receptors Synergizes the Effect of Systemic Ibuprofen in a Pain Model in Rat

**DOI:** 10.3390/ph15080910

**Published:** 2022-07-23

**Authors:** M. Irene Díaz-Reval, Yolitzy Cárdenas, Miguel Huerta, Xóchitl Trujillo, Enrique Alejandro Sánchez-Pastor, María Eva González-Trujano, Adolfo Virgen-Ortíz, M. Gicela Pérez-Hernández

**Affiliations:** 1Centro Universitario de Investigaciones Biomédicas, Universidad de Colima, Colima 28045, Mexico; rosa_cardenas@ucol.mx (Y.C.); huertam@ucol.mx (M.H.); rosio@ucol.mx (X.T.); espastor@ucol.mx (E.A.S.-P.); avirgen@ucol.mx (A.V.-O.); 2Laboratorio de Neurofarmacología de Productos Naturales, Dirección de Investigaciones en Neurociencias, Instituto Nacional de Psiquiatría Ramón de la Fuente Muñiz, Ciudad de Mexico 14370, Mexico; evag@imp.edu.mx; 3Facultad de Enfermería, Universidad de Colima, Colima 28045, Mexico; ggise_ph@ucol.mx

**Keywords:** synergism, antinociception, inflammatory pain, formalin model, cannabinoids receptors, ibuprofen, WIN 55,212-2, rats

## Abstract

Pharmacological synergism is a current strategy for the treatment of pain. However, few studies have been explored to provide evidence of the possible synergism between a non-steroidal anti-inflammatory drug (NSAID) and a cannabinoid agonist, in order to establish which combinations might be effective to manage pain. The aim of this study was to explore the synergism between ibuprofen (IBU) and the synthetic cannabinoid WIN 55,212-2 (WIN) to improve pain relief by analyzing the degree of participation of the CB_1_ and CB_2_ cannabinoid receptors in the possible antinociceptive synergism using an experimental model of pain in Wistar rats. First, the effective dose thirty (ED_30_) of IBU (10, 40, 80, and 160 mg/kg, subcutaneous) and WIN (3, 10, and 30 µg/p, intraplantar) were evaluated in the formalin test. Then, the constant ratio method was used to calculate the doses of IBU and WIN to be administered in combination (COMB) to determine the possible synergism using the isobolographic method. The participation of the CB_1_ and CB_2_ receptors was explored in the presence of the antagonists AM281 and AM630, respectively. The combination of these drugs produced a supra-additive response with an interaction index of 0.13. In addition, AM281 and AM630 antagonists reversed the synergistic effect in 45% and 76%, respectively, suggesting that both cannabinoid receptors are involved in this synergism, with peripheral receptors playing a relevant role. In conclusion, the combination of IBU + WIN synergism is mainly mediated by the participation of the CB_2_ receptor, which can be a good option for the better management of pain relief.

## 1. Introduction

Drug combinations are a therapeutic strategy for improving pharmacological efficacy. Concerning pain, different combinations have been developed for its treatment [1,2]. The study of synergism has resulted in the development of drug combinations to improve the analgesic efficacy and, at the same time, to reduce the drug doses to avoid severe side effects in patients with chronic treatment. Therefore, supra-additive synergism will be ideal for the achievement of this aim. This kind of synergism can be obtained with drugs of different action mechanisms, or with drugs administered by different routes. For example, Pozos-Guillen et al. [3] reported that intraplantar and intraperitoneal injections of tramadol produced self-synergism. The authors suggest that tramadol activates different mechanisms related to the administration route [3]. Additionally, a self-synergism was reported with WIN 55,212-2 administered in two different routes, the intrathecal and intraplantar [4]. However, other studies have only shown an additive effect [5], or there was no change in the antinociceptive effect with the administration of tramadol by two different routes [6].

Non-steroidal anti-inflammatory drugs (NSAIDs) are used for mild to moderate pain management, being especially helpful in inflammatory pain. Ibuprofen is an analgesic derived from propionic acid that has been on the market for 50 years. It is a widely used over-the-counter drug; there are forecasts that its market will grow by 6.1% in the next 5 years [2]. In the clinic, it is used for rheumatoid arthritis, osteoarthritis, and the treatment of other pathologies globally because it is more effective and safer than other drugs of the same family. However, ibuprofen also induces gastrointestinal and cardiovascular side effects. Nevertheless, these effects are less frequent than those of aspirin [7,8].

The traditional medicine of some cultures has utilized cannabinoids of vegetal origin to alleviate pain for a long time [9]. The empiric use of these plants is based on the alcoholic maceration of vegetal material. After that, the mixture is placed in the zone where the patient indicates that they feel pain. Some studies have reported the contribution of CB_1_ and CB_2_ cannabinoid receptors in the nociceptive signal inhibition of different kinds of pain including nociceptive and inflammatory pain [10,11,12].

CB_1_ receptors are located in several areas of the central nervous system (CNS) and are involved in the endogenous analgesic pathway. These receptors not only participate in analgesia, but also modulate other functions including the psychotropic effects of cannabinoids when administered by the systemic route [13], which are the most severe side effects. Additionally, the presence of these receptors in nociceptors has been reported. This suggests that their contribution to antinociception is more significant in the nociceptors than in the CNS [14].

Some studies have demonstrated the localization of CB_2_ receptors in the immune system [12], in healthy and damaged human dorsal root ganglion (DRG) sensory neurons, and in nervous peripheric fibers [15]. CB_2_ selective agonists can inhibit nociception, while selective antagonists revert the antinociceptive effect [15,16].

Vegetal origin and synthetic cannabinoids have shown antinociceptive effects in diverse pain models. However, few studies on the synergism between NSAIDs and cannabinoids have explored their efficacy in pain management. The intraplantar injection of ibuprofen or rofecoxib combined with anandamide yielded supra-additive synergism [17,18], while the systemic administration of WIN 55,212-2 plus ketorolac produced additive synergism [19]. Drugs such as selective COX2 inhibitors have also shown a synergistic effect when co-administered with cannabinoids [20]. 

It is widely reported that the oral route is the most common route of administration for NSAIDs, and cannabinoids induce side effects in the CNS when administered by the systemic route. The aim of this study was to analyze the possible synergism between IBU and WIN, identifying the percentage of participation of the CB_1_ and CB_2_ cannabinoid receptors. This combination might be a good option for patients suffering from chronic nociceptive and inflammatory pain using oral and topic administration, respectively, since both administration routes are comfortable, and have the advantage of using cannabinoids at low doses, reducing the adverse effects.

## 2. Results

### 2.1. Evaluation of the Antinociceptive Effect

To induce nociception, an intraplantar (i.pl.) administration of 5% formalin was used in rats. This model is useful for screening mild analgesics by producing a biphasic response. The first phase corresponds to the early nociception generated by direct action on nociceptors. The second phase represents the late nociception generated by the release of inflammatory mediators. In this study, IBU was tested at different subcutaneous (s.c.) doses, and did not show an antinociceptive effect in the first phase. Therefore, antinociception corresponded to the second phase of the test. Figure 1 shows that the initial effect was elicited at a dosage of 10 mg/kg, while the maximum efficacy (56.6 ± 2.52%) was obtained at a dose of 160 mg/kg.

On the other hand, WIN, a non-selective CB_1_ and CB_2_ cannabinoid receptor agonist, was locally (i.pl) administered at the same site as formalin, in doses of micrograms. The lowest dosage (3 µg/i.pl.) produced an effect of 31.15 ± 2.81%, 10 µg/i.pl. produced a 46.15 ± 1.87% effect, and the highest dosage (30 µg/i.pl.) did not significantly modify the maximal response (50.79 ± 3.05%; *p* > 0.05).

The ED_30_ of IBU and WIN was calculated from the DRCs of each drug, which are shown in Table 1. According to these results, WIN was more potent and less effective than IBU. The doses of both drugs to be combined were obtained from the ED_30_, as determined from the DRCs, as explained in the experimental protocol.

The DRC of the drug combination was constructed with four doses. Figure 1 shows the DRC of the combination of IBU + WIN (COMB). Under these experimental conditions, the efficacy of COMB was similar to that of IBU. However, COMB was more effective than WIN alone. It is relevant to highlight that much lower doses of both drugs were used in the DRC of COMB. Therefore, the DRC of COMB shifted to the left compared to the DRC of ibuprofen. This shift indicates that an increase in potency was produced in the coadministration of drugs. Thus, the ED_30_ of the combination was significantly lower than that of IBU (*p* < 0.05, Table 1).

Figure 2 shows the doses of the drugs that, in combination, presented the maximal efficacy. In a single administration, 39.54 mg/kg of IBU and 2.74 µg/i.pl. of WIN showed an effect of 28.99 ± 2.21% and 39.55 ± 1.71%, respectively. The co-administration of both drugs gave an effect of 63.56 ± 2.14%. This effect was statistically different compared to IBU (*p* < 0.0001) and WIN ipsi (*p* < 0.01). What is interesting about this combination is the dose reduction in both drugs. The maximal efficacy of IBU (56.6 ± 2.52%) was obtained with a dosage of 160 mg/kg, and with the combination, a similar efficacy was achieved, but with a 4-fold lower dose of IBU. In the case of WIN, it was reduced 3.6-fold. Since WIN produces CNS effects, an independent group of rats was administered with WIN on the contralateral paw (WIN cont) at a dosage of 10 µg/i.pl. Compared with the control group, no statistically different effect was observed in this group. However, there was a significant difference compared with the WIN ipsi group (*p* < 0.01). These data demonstrate a local effect at this dose and when administered in combination with IBU.

### 2.2. Determination of Synergism

Isobolographic analysis was performed to determine the synergism generated by COMB. First, the value of ED_30_ for IBU (s.c.) and WIN (i.pl.) were plotted, then these points were joined with a straight line (Figure 3), where the letter T shows the theoretical ED_30_. The experimental results showed that a 30% effect was obtained in the combination of 2.75 mg/kg of IBU and 0.01 µg/i.pl. of WIN. This dose combination was located below the isobola and was far from the theoretical ED_30_. It can be identified with the letter E. The statistical analysis showed a statistical difference between both doses (*p* < 0.0001) with an interaction index (γ) = 0.13, indicating a supra-additive synergism. 

### 2.3. Involvement of CB_1_ and CB_2_ Receptors in Antinociceptive Synergism

The supra-additive antinociceptive synergism obtained with the combination of IBU and WIN demonstrated that the minimal dose of local administration of this cannabinoid significantly increased the efficacy of the systemic administration of ibuprofen, which led us to analyze the participation of cannabinoid receptors in this synergism. Selective antagonists to the CB_1_ (AM281) and CB_2_ (AM630) receptors administered locally and prior to the most effective combination (IBU 39.54 mg/kg and WIN 2.74 µg/i.pl) were investigated.

The synergistic effect of COMB (63.56 ± 2.14%) was significantly reduced at 45% (*p* < 0.0001) in the presence of AM281 in the formalin test (Figure 4), suggesting the participation of the CB_1_ receptor in the antinociceptive synergism. On the other hand, the synergism of COMB was significantly reduced at 76% (*p* < 0.0001) in the presence of AM630. These results suggest that in comparison to CB_1_ receptors, CB_2_ receptors are mainly involved in the synergism produced with COMB. A third and independent group of rats was administered with both antagonists before the COMB treatment. Thus, the antinociceptive effect was diminished at −21%, supporting the participation of both the CB_1_ and CB_2_ cannabinoid receptors and their interaction with other inhibitory neurotransmission systems in the synergism generated by the COMB (IBU + WIN).

## 3. Discussion

Previous studies have reported that the cannabinoid system is involved in pain inhibition, where CB_1_ receptors are found in both presynaptic and postsynaptic terminals in the CNS, while CB_2_ receptors are mainly located in the microglia and postsynaptic cells, identified in neuronal damage and inflammation [21]. In the peripheral areas, CB_1_ receptors are located on nociceptors and CB_2_ receptors in immune system cells involved in inflammation [22]. In therapy, the systemic administration of cannabinoids represents a clinical problem due to the adverse effects originating at the central level. Studies have reported that these undesirable effects depend on the activation of CB_1_ receptors [23]. Therefore, a possible strategy for the treatment of pain to reduce these adverse effects is local administration.

It is known that the CB_1_ and CB_2_ receptors can be activated in pain diseases such as arthritis, gout, and musculoskeletal trauma. WIN 55,212-2 possesses a high affinity for both receptors [24]. In the present study, this non-selective agonist was administered to rats by the intraplantar route in the same extremity where nociception was generated to explore cannabinoid participation. Local injection was used to activate only peripheral cannabinoid receptors, avoiding their action in the CNS to prevent psychotropic effects. In our study, a maximal efficacy of 46.15 ± 1.87% was obtained after the administration of WIN. This result agrees with previous studies using the formalin test, where intrathecal administration showed the maximal efficacy of this drug in the inflammatory phase of approximately 50% [25]. In acute pain induced by thermal stimulus, a maximal efficacy of 34.6% was obtained after the intraperitoneal administration of WIN [26]. Meanwhile, for cancer pain, it has been reported that spontaneous activity decreases in C-fibers due to the activation of both the CB_1_ and CB_2_ receptors [22].

The great usefulness of ibuprofen in different pain states, due to the low incidence of adverse effects, has led us to think that there is a high probability that it can be co-administered with cannabinoids. Therefore, in this study, we evaluated different doses in monotherapy and combined with WIN and observed that it produced a maximal effect of 56.6 ± 2.52% using 5% formalin as a nociceptive agent, as previously reported in other studies [27]. The IBU + WIN combination reached a maximal efficacy of 63.56 ± 2.14%, demonstrating a greater effect than IBU alone. Doses of individual drugs were significantly reduced when combined, indicating that antinociceptive potency was increased. The administration of low doses may subsequently result in a possible decrease in adverse effects; thus, patients who undergo chronic treatment with analgesics might use this therapy for longer periods.

The isobolographic analysis showed that COMB generated a potentiation synergism with an interaction index of 0.13; according to Tallarida [28], an interaction index of less than one indicates supra-additive synergism. This finding agrees with data from Guindon et al. [17], who reported a synergistic effect when combining IBU and anandamide, an endogenous cannabinoid, using local administration. In the present study, IBU was administered systemically to involve the peripheral and central mechanisms. In addition to the inhibition of prostaglandin synthesis at the peripheral level, IBU can also stop synthesis in the CNS. It has also been reported that when administered at the same site, IBU can inhibit (nitric oxide) NO synthesis [29] and decrease the nociceptive effects generated by glutamate and substance P [30], contributing to its analgesic activity. In our study, WIN was administered via the intraplantar route to activate only local cannabinoid receptors, suggesting that different mechanisms probably contribute to the observed supra-additive synergism (Figure 5).

The degree of involvement of the cannabinoid receptors observed in the supra-additive synergism for pain inhibition was explored in this study, and differences were found with previous reports such as the results of Guindon et al. [17], who reported that the CB_1_ receptor participated more strongly in the synergism observed with IBU and anandamide. In this respect, we observed that the AM281 antagonist reduced the effect of the combination at 45%, while the AM630 antagonist produced a reduction of 76%. Thus, the CB_2_ receptor showed stronger participation in the synergism of our COMB with these experimental conditions. Other results agree that the CB_2_ receptor agonists administered via the intraplantar route have similar efficacy to morphine by the same route [21].

In addition, it is known that the CB_2_ receptor is expressed in leukocytes, neutrophils, monocytes, and T cells, among other cells involved in inflammation [12]). It has also been reported to play an essential role in controlling the acute inflammatory response [31]. This effect has been explained in some experimental models because the activation of this receptor can reduce neutrophil recruitment [32] and decrease IL-6 secretion [24], resulting in anti-inflammatory effects. Thus, the activation of CB_2_ receptors could inhibit the release of inflammatory and pain mediators in the late phase of the formalin test. Together with the inhibition of prostaglandin synthesis induced by ibuprofen, this information supports the synergistic response of COMB in our study (Figure 5).

With regard to the CB_1_ receptors that are expressed in nociceptors, Agarwal et al. [14] reported that these receptors play a significant role in analgesia since these receptors induce endocannabinoid release both peripherally and centrally. Moreover, small doses of synthetic cannabinoid agonists reduce inflammatory and neuropathic pain. In our study, low doses of WIN administered in the hind paw produced supra-additive synergism in which the CB_1_ receptor showed weaker participation than CB_2_.

The effect of COMB was avoided completely when the antagonists of the CB_1_ and CB_2_ receptors were tested together, since animals presented a higher number of nociceptive behaviors than the control group. This result demonstrates the synergistic role of both cannabinoid receptors in the antinociceptive effect of the combination. It is known that the activation of the CB_1_ and CB_2_ cannabinoid receptors by WIN inhibits the activity of TRPV1 receptors [33,34], which are involved in several types of pain including those associated with inflammatory pain induced by chemicals such as formalin. Therefore, the synergism obtained with the COMB treatment might be in part due to the activation of the CB_2_ receptors in the inflammatory cells by WIN, which leads to the inhibition of inflammatory mediators such as interleukins and cytokines so that these mediators will no longer desensitize the nociceptor. On the other hand, WIN also activates the CB_1_ receptor located on nociceptors, contributing to the inhibition of the nociceptive signal (Figure 5). In addition, endocannabinoids are released under stress stimuli such as inflammation and pain [35]. Calignano et al. [36] reported that the levels of the endocannabinoids anandamide and palmitoylethanolamide in rat paw skin could activate the CB_1_ and CB_2_ receptors, respectively, attenuating nociceptive behaviors. Therefore, when both the CB_1_ and CB_2_ receptor antagonists were co-administered, the nociceptive effect was higher than in the control group. These mechanisms, added to the peripheral inhibition of prostaglandins and the central mechanisms activated by IBU, induced the synergism observed in our experimental design.

There are studies that have demonstrated the participation of the cannabinoid system in analgesia [14,21] as well as other studies that have found synergism or addition with combinations of NSAIDs and cannabinoids [1,37]. However, there is a need for studies to determine the dosage, safety, and administration conditions. Our study demonstrated that synergism occurred with the coadministration of IBU plus WIN when administered by different routes. The administration of IBU by the systemic route activates peripheral and central mechanisms, whereas the administration of WIN by the local route activates only peripheral mechanisms, avoiding adverse effects in the CNS.

The limitation of this study is that no adverse effects were analyzed. Further studies are needed to determine if this combination might be associated with possible adverse effects. Nevertheless, our results provide evidence of the potential benefit of this combination for pain therapy, and suggest that a formulation integrating a non-selective cannabinoid with a NSAID such as IBU might be useful for pain relief.

## 4. Materials and Methods

### 4.1. Animals

Male Wistar rats (180–220 g in weight, age 6–8 weeks) were purchased from our vivarium. The animals were housed with a 12 h light/12 h darkness cycle, at a relative humidity of 40–60% and a controlled temperature (23 ± 2 °C), with free access to water and food. One day before the experiment, the rats were taken to the laboratory to adapt to the ambient conditions and for manipulation. The laboratory was maintained at temperature and humidity conditions similar to those of the vivarium. The food was removed 12 h before the experiment. All experiments were conducted following International Ethical Standards for Pain Studies in Animals [38], and the Official Mexican Standard NOM-062-ZOO-1999 technical specifications for the care and use of laboratory animals. The protocol was approved by the Bioethics Committee of Centro Universitario de Investigaciones Biomédicas, Universidad de Colima (Project code 2020-11; approval date: 30 April 2020).

### 4.2. Drugs and Solutions

WIN 55,212-2 (cat. 1038), a no selective agonist of the CB_1_ and CB_2_ cannabinoid receptors, and AM281 (cat. 1115) and AM630 (cat. 1120) as selective antagonists to the CB_1_ and CB_2_ receptors, respectively, were obtained from Tocris Cookson Inc., Ellisville, MO, USA. Ibuprofen (cat. I1892), a nonsteroidal anti-inflammatory analgesic, NaCl (cat. S7653), Tween 80, and DMSO (cat. D5879) used to prepare the vehicles were purchased from Sigma-Aldrich Laboratory, Toluca, México.

A stock solution with a concentration of 1 µg/µL was prepared for each of the cannabinoids in 5% DMSO plus 5% Tween 80 and 90% physiological saline. Subsequently, each dose was administered in a volume of 50 µL [22].

### 4.3. Formalin Model

Nociception was induced by using the formalin test in rats [27]. Briefly, rodents were placed in a chamber for observation. The camera consisted of a Plexiglas cylinder 30 cm high and 20 cm in diameter. Two mirrors were placed at the back of the cylinder for a total overview of the animals. The rats were weighed and placed in the chamber for 20 min to allow them to adapt to their new environment. Subsequently, drugs were administered according to the treatment group. Then, 50 µL of 5% formalin i.pl. was administered in the right hind paw. Paw flinches were recorded as nociceptive behavior. A recording was taken every 5 min for a total period of 60 min. At the end of each experiment, animals were euthanized by cervical dislocation.

### 4.4. Experimental Protocol

The control group received the solutions used to dissolve the experimental drugs. One group of six rats received a subcutaneous (s.c.) injection of physiological saline, and five minutes later, 50 µL of a mixture of 90% physiological saline, 5% DMSO, and 5% Tween 80 was administered via the intraplantar (i.pl.) route. Ten minutes later, 50 µL of 5% formalin was administered. Immediately, the rats were placed in the observation chamber. The nociceptive behaviors were recorded. These solutions, administered by different routes, were the vehicles in which the IBU and cannabinoids were dissolved, respectively.

The next group was injected with the analgesic IBU (s.c.) at 10, 40, 80, and 160 mg/kg doses. Then, another group of rats was treated with the cannabinoid WIN (i.pl.) in the same extremity injected with formalin. The doses used were 3, 10, and 30 µg/p [25]. In addition, the local effect of WIN was evaluated with the administration of a dose of 10 µg/p in the contralateral paw. Finally, the COMB was administered as indicated in Table 2. The drugs were supplied according to the times indicated for the controls, each dose having *n*= 6.

The selective antagonists AM281 and AM630 were explored. This administration evaluated the degree of participation of each of the cannabinoid receptors in the analgesic synergism. The drug combination used was the one that produced the maximal efficacy, 39.54 mg/kg IBU + 2.74 µg/p WIN. To one group of six rats was administered AM281 (i.pl.), a CB_1_ receptor antagonist, in a volume of 50 µL at a dose of 0.3 µg/p. Then, 5 min later, IBU s.c.; another 5 min later, WIN i.pl.; and at the end of 10 min, formalin [22,34]. Another six rats underwent similar administration, but the CB_2_ receptor antagonist, AM630 (i.pl.), at a dose of 3 µg/p, was used [22,34]. Finally, a third group was administered the two antagonists simultaneously in a volume of 50 µL (i.pl.), and subsequently, the maximally effective combination. For each group, *n* = 6.

### 4.5. Data Analysis

Regarding the ethical principles for the study of pain in animals, the minimum number of rats to obtain significant results for each dose in the different dose–response curves was six individuals. The type of sampling used was simple random.

For the calculation of the sample size in animals, the resource equation was used, which implements the principles of the 3Rs (Reduction, Refinement, and Replacement), in order to determine the minimum sample size necessary to detect the probability of an error in an analysis of variance.

The minimum number of animals per group (*mnapg*):(1)$$mnapg=\left(\frac{10}{K}\right)+1$$

The maximum number of animals per group (*MNAPG*):(2)$$MNAPG=\left(\frac{20}{K}\right)+1$$
where *K* is the number of treatments, and 10 and 20 are the minimum and maximum degrees of freedom, respectively, for a one-way ANOVA [39,40].

Therefore, different treatments are described in the experiment. Each treatment used a different dose, so the calculation was performed individually depending on the dose used in each treatment. The total number of rats for the whole study was 108. Accordingly, all results are expressed as the mean ± SEM for six animals per group.

Nociceptive behaviors were recorded every 5 min for a total period of 60 min, from which time courses (TC) were constructed. Only phase two of the model was analyzed in this study. Subsequently, the cumulative nociceptive effect was calculated as the area under the curve (AUC) by the trapezoidal method. This value was used to calculate the percentage of antinociception according to the following formula:(3)$$\%\;Antinociception=\left[\frac{\left(AUC_{C}-AUC_{D}\right)}{AUC_{c}}\right]\times 100$$
where *AUC_C_* is obtained from the control group, and *AUC_D_* is obtained from the treated group in the presence of drugs.

Isobolographic analysis is a suitable tool for analyzing the interactions between analgesics [41,42]. Therefore, in this study, synergism was determined with an isobologram. This method is based on the use of equipotent doses of the drugs. Generally, the analysis uses the ED_50_ of each drug. However, other doses can be used. In this study, the ED_30_ was calculated due to the maximal effect presented by WIN. Subsequently, the DRC of the COMB was constructed with the ED_30_ of each drug, as shown in Table 2. A constant dose ratio (fixed ratio) was maintained with this method. Finally, the ED_30_ of the COMB was determined.

The Isobologram was constructed as follows: The ED_30_ of IBU is shown on the X-axis. On the Y-axis, the ED_30_ of WIN is located. The oblique line joining the two ED_30s_ is the additive theoretical line. The center point of the line, indicated by T, is the theoretical ED_30_. The point marked E is the experimental ED_30_ calculated from the DRC of COMB

The interaction index (*γ*) was calculated to determine the type of synergism that occurs with the interaction of two drugs. For this purpose, the theoretical *ED_30_* of the COMB was obtained. This dose corresponds to the sum of the effects of both drugs administered individually. The interaction index was calculated with the following formula:(4)$$\gamma =\frac{ED_{30}Experimental}{ED_{30}Theoretical}$$

The interaction index indicates what part of the ED_30_ of the individual drug represents the corresponding ED_30_ in the combination. The value of this index indicates the type of synergism present in a drug interaction. Thus, a value close to 1 corresponds to an additive synergism, a value greater than 1 corresponds to an infra-additive synergism, and a value less than one corresponds to a supra-additive synergism [42].

### 4.6. Statistical Analysis

Data are shown as the mean ± standard error of the mean (SEM) of *n* = 6 repetitions for each treatment. Linear regression by the least-squares method was used to determine the effective dose 30 (ED_30_).

One-way analysis of variance (ANOVA) followed by a post hoc Tukey test was performed to compare the percentages of the antinociceptive effect of each treatment.

The isobologram calculations were performed using the method reported by Tallarida [42]. The statistical difference between the theoretical and experimental doses was calculated with a Student’s *t*-test for independent means. A value of *p* < 0.05 was considered significant.

Statistical analysis, the determination of ED_30_, and the creation of graphics were performed with Graph Pad Prisma for Windows version 6 (GraphPad Software, San Diego, CA, USA).

## 5. Conclusions

In conclusion, the combination of IBU + WIN produced a significant antinociceptive synergistic effect in which the CB_2_ cannabinoid receptors were mainly involved compared to the participation of the CB_1_ receptors. Therefore, the combination of NSAIDs and cannabinoids administered systemically and locally, respectively, may be an option in treating inflammatory pain.

## Figures and Tables

**Figure 1 pharmaceuticals-15-00910-f001:**
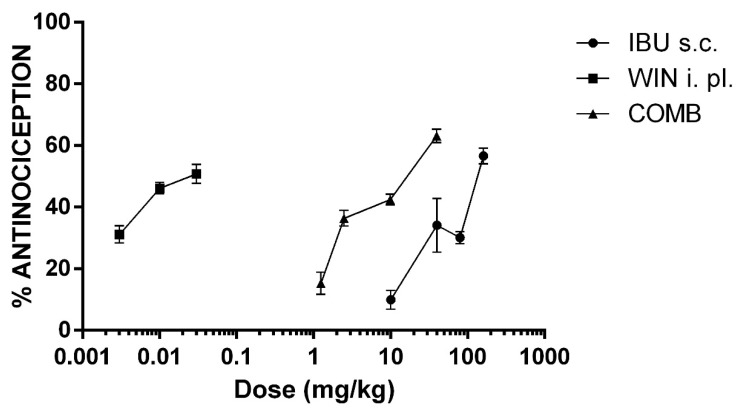
The dose–response curves (DRC) of the antinociceptive effect of ibuprofen (IBU, 10, 40, 80, and 160 mg/kg, s.c.), WIN 55,212-2 (WIN, 3, 10, and 30 µg/i.pl.), and the combination of both drugs (COMB). For COMB, each drug was administered by their respective route. The mean ± the SEM is plotted with six repetitions for each dose.

**Figure 2 pharmaceuticals-15-00910-f002:**
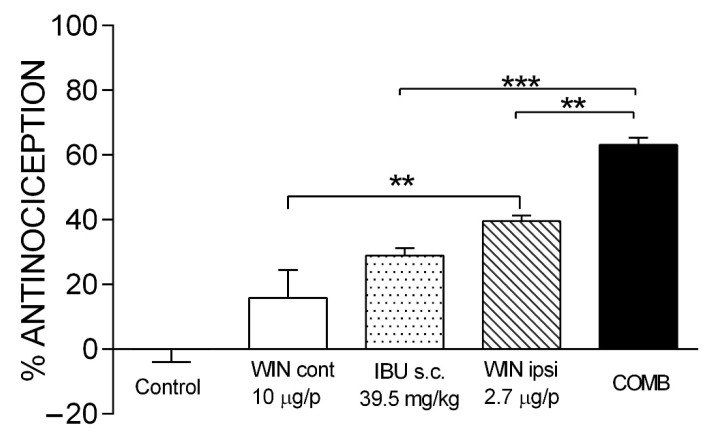
The antinociceptive effect of maximal efficacy COMB. The individual effects of IBU and WIN are shown as well as those of COMB. WIN was administered on both the ipsilateral paw (WIN ipsi) and contralateral paw (WIN cont) in the formalin model. There was no statistically significant difference between WIN cont and the control. Groups showing significant differences after one-way ANOVA were WIN ipsi compared with WIN cont (** *p* < 0.01), COMB compared with IBU (*** *p* < 0.001), and COMB compared with WIN ipsi (** *p* < 0.01). Each bar corresponds to the mean of six repetitions ± SEM.

**Figure 3 pharmaceuticals-15-00910-f003:**
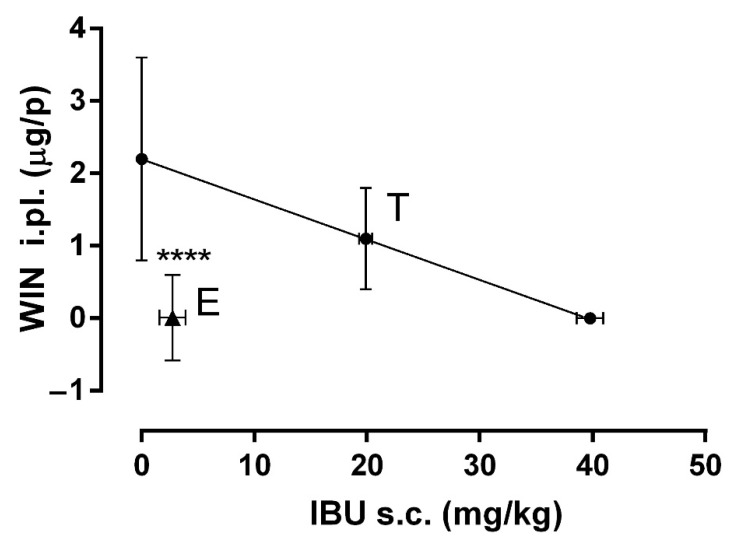
The isobologram showing the interaction between IBU (s.c.) and WIN (i.pl.) in the formalin model. The ED_30_ of IBU is shown on the X-axis and the ED_30_ of WIN on the Y-axis. The oblique line joining the two ED_30_ is the additive theoretical line. The center point of the line indicated by T is the theoretical ED_30_. The point marked as E is the experimental ED_30_ calculated from the DRC of COMB. A significant difference between points T and E indicate a supra-additive synergism (**** *p* < 0.0001). Data are shown as the mean ± SEM of six repetitions.

**Figure 4 pharmaceuticals-15-00910-f004:**
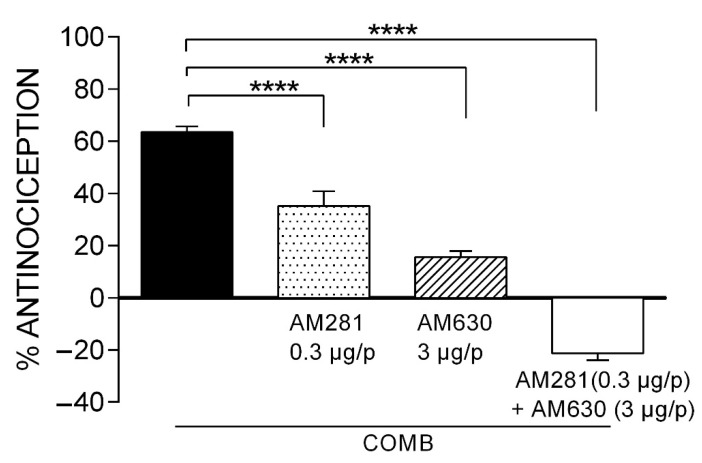
The effect of the CB_1_ and CB_2_ receptor antagonists on the synergism induced by the combination of IBU (39.54 mg/kg, s.c.) and WIN (2.74 µg/p, i.pl.). Statistical significance is indicated between the COMB group and groups receiving the individual antagonists or their combination (**** *p* < 0.0001). Data are expressed as the mean ± SEM of six repetitions.

**Figure 5 pharmaceuticals-15-00910-f005:**
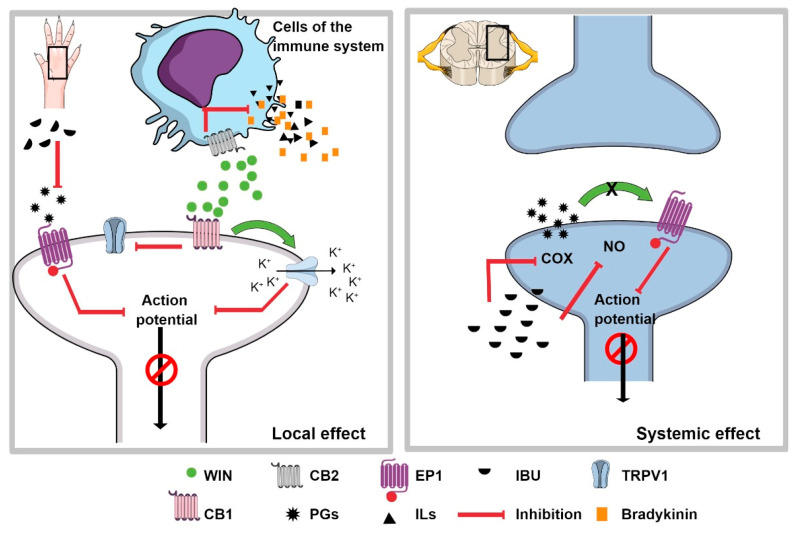
The synergism of IBU + WIN. The left side shows the possible mechanism of the local effect of WIN and IBU, the latter representing the inhibition of the synthesis of prostaglandins (PGs), which causes the non-sensitization of the nociceptor. In addition, when the CB2 receptor, located on immune system cells, is activated by WIN, it inhibits the release of pain and inflammatory mediators, which prevents nociceptor sensitization. When the CB1 receptor interacts with WIN, the βγ-dimer can open potassium channels to hyperpolarize the nociceptor. In addition, it can also inhibit the TRPV1 receptor. All of these events inhibit the nociceptive signal. The right inset represents the possible mechanism at the spinal level, where systemically administered IBU can inhibit the synthesis of PGs and nitric oxide (NO), which centrally inhibits the transmission of the nociceptive signal. These mechanisms would explain the synergism observed with the combination IBU + WIN administered systemically and locally, respectively.

**Table 1 pharmaceuticals-15-00910-t001:** The effective dose 30 (ED_30_) of the drugs in the study, each administered through the corresponding route.

Treatment	ED_30_	Route of Administration
IBU	39.54 mg/kg	s.c.
WIN	2.74 µg/p	i.pl.
COMB	2.76 mg/kg	s.c. + i.pl.

**Table 2 pharmaceuticals-15-00910-t002:** The doses used in the study to analyze the synergism between IBU + WIN, each administered by the corresponding route corresponding.

	IBU s.c. mg/kg	WIN i.pl. µg/p	COMB Total Dose
ED_30_	39.54	2.74	39.54274
ED_30_/4	9.89	0.69	9.89070
ED_30_/16	2.48	0.18	2.48020
ED_30_/32	1.24	0.09	1.24009

## Data Availability

Data is contained within the article.

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
