# Peer review of "Activation of Peripheral Cannabinoid Receptors Synergizes the Effect of Systemic Ibuprofen in a Pain Model in Rat"

_pharmaceuticals, 2022, doi:10.3390/ph15080910_

Round 1

Reviewer 1 Report

The authors have improved what was suggested. The paper is interesting but further experiments have to be done in the future to explain better and characterize the synergistic effect and/or the adverse effects.

Author Response

The authors thank you for your valuable time spent reviewing the manuscript, which has helped us to improve it. We have improved each of the sections of the manuscript which gives it a higher quality. A limitation of our study is that adverse effects were not part of our objective. In another type of study this issue can be analyzed. 

Reviewer 2 Report

This work studied the synergistic effect added by activation of peripheral cannabinoid receptors to the effect of systemic ibuprofen in a rat model of pain. There are some issues in this manuscript that should be addressed as follows:

·         Title: The word “synergize” should be replaced with “synergizes”.

Title: The word “cannabinoids” should be replaced with “cannabinoid”.

·         Abstract Line 21: The sentence “experimental model pain” should be replaced with “experimental model of pain”.

·         The meaning of the abbreviations should be clearly defined at their first mention (e.g. NSAID).

·         Keywords: The word “rats” should be added to the keywords.

·         The novel points in this study should be clarified because there are previous reports that discussed a similar topic, e.g.https://www.ncbi.nlm.nih.gov/pmc/articles/PMC3056416/, https://www.ncbi.nlm.nih.gov/pmc/articles/PMC7590544/

·  The first and the last paragraphs of the “Introduction” should be enriched with more recent references.

  • The exact source, concentrations and the catalogue numbers of the used kits and chemicals should be mentioned.
  • How did you know that the animals were acclimatized?
  • The exact number of rats used in this study should be mentioned.
  • Why did the authors choose ibuprofen among NSAIDs?
  • References for the used doses of drugs and duration of treatment should be added.
  • Page 10 Line 405: The subheading “Statistically Analysis” should be replaced with “Statistical Analysis”.
  • A collective diagram summarizing the main findings of this study is recommended.
  • The discussion should provide more details to analyze of the results of the present study.
  • The manuscript should be revised by English-naïve speaker to improve the quality of the language.
  • The manuscript should be checked regarding the grammatical errors and plagiarism.
